# Factors Affecting Accountants' Adoption of Remote Working: Evidence from Jordanian Governmental Organizations



**Qutaiba Adeeb Odat** [1]**, Hashem Alshurafat** [1]**, Mohannad Obeid Al Shbail** [2]**, Husam Ananzeh** [3] **and Hamzeh Al Amosh** [4,5,*]

[1] Department of Accounting, Business School, The Hashemite University, P.O. Box 330127, Zarqa 13133, Jordan
[2] Department of Accounting, School of Business, Al Al-Bayt University, Mafraq 25113, Jordan
[3] Department of Accounting, Irbid National University, Irbid 21110, Jordan
[4] Ministry of Education and Higher Education, Doha 35111, Qatar
[5] Faculty of Business and Management, University Sultan Zainal Abidin, Campus Gong Badak, Kuala Terengganu 21300, Malaysia
[*] Correspondence: hamza_omosh@yahoo.com

**Abstract:** This study aims to examine the factors that impact accountants' adoption of remote working within Jordanian governmental organizations. Several models have proven to explain the acceptance and usefulness of technology. Therefore, this study provides an integrated model from a set of theories, including the Technology Acceptance Model (TAM), Technology–Organization–Environment (TOE), Social Capital Theory (SCT), and Theory of Reasoned Action (TRA). The data were gathered from 155 accountants working in Jordanian governmental organizations. PLS-SEM was performed on the data using the Smart-PLS 3 software. The study found a positive relationship between organizational culture, management support, policies, perceived ease of use, and subjective norms with perceived usefulness. In addition, the study found a positive supported relationship between perceived ease of use, perceived usefulness, and subjective norms and attitude toward use. Lastly, this study found a positive supported relationship between attitude toward use and behavioural intention and actual use. This study recommends that future research expand the model by adopting the technological context. In addition, further research could implement the study over a new geographical context in developed and developing countries and compare outcomes.

**Keywords:** accounting profession; remote working; governmental organizations; Jordan

## 1. Introduction

Many rules are based on restrictions in international accounting careers impacted by the COVID-19 pandemic [1]. As the gradually increasing virus spread, nations took strict measures and procedures to prevent chaos, such as the social distancing that led to remote working. These sudden procedures might shock many organizations with no risk or crisis plans for the acceptance of technology by employees, especially accountants. The tragic pandemic has impacted many areas, especially the business and economic environment [2]. Accordingly, most countries adopted restricting social connection as an essential plan to flatten the infection curve during the pandemic [3]. Consistently strict rules were implemented regarding working conditions, which obligated most companies and enterprises to switch to working from home. Converting most jobs into remote working was accelerated to help companies sustain their operations.

Employees such as accountants can adapt to changes and respond to challenges influenced by factors such as the length of the adjustment period and the degree of instructions they receive. When daily life is disrupted, immediate actions, such as work routines, are implemented, and a sense of responsibility is necessary [3,4]. Remote working is not new; however, difficult times such as COVID-19 increased organizations' attention

to remote working. Thus, accountants were obligated to work remotely during the pandemic, with social restrictions and distance policies [1]. Therefore, this study aims to examine the factors that impact accountants' adoption of remote working within Jordanian governmental organizations.

Several studies have dealt with technology acceptance through structured theories and models to explore the possible factors that have significant influences and implications [5–12]. This study examines accountants' perceptions of the factors that impact their adoption of remote working within Jordanian governmental organizations. A developed model has been designed to support accountants in dealing with the impacts of the lockdown on their jobs, including proposed influential factors on users' intent to accept new accounting technologies. This study employs several theoretical models, such as the TAM, TOE, SCT, and TRA, to interpret the factors that, accordingly, influence accountants' overall acceptance and use of remote technology within Jordanian governmental organizations.

This study's problem is to investigate several concerns, such as how the COVID-19 epidemic has affected the accounting profession in Jordan and how government accountants have attempted to adjust to the enforcing of the emergency procedures. Moreover, the issues that have arisen are more likely to impact accountants' efficiency in their professional activities. Therefore, this study answers the following question:

What factors affect remote working acceptance and use by accountants within Jordanian governmental organizations?

This study's findings indicate a positive correlation between organizational culture, management support, policies, ease of use, subjective norms, and the perceived usefulness of a technology. Furthermore, a positive correlation exists between ease of use, perceived usefulness, subjective norms, and users' attitudes towards technology. Additionally, the study found a positive correlation between users' attitudes and their intentions to use technology and their actual usage. This study brings significant contributions to both theory and practice. Firstly, it motivates existing accounting research to consider a wider range of factors when studying remote working adoption. The findings of this study suggest that the factors included in this study have impacted remote working during the COVID-19 pandemic. As a result, this study expands our understanding of the remote working adoption process.

This study is important as it provides insight into the perceptions and attitudes of accountants towards remote working and identifies the factors that influence their adoption of this work arrangement. This information can be used by organizations in Jordan and other similar contexts to develop strategies and policies to support the adoption of remote working among their employees. The study can inform the development of training programs and the provision of technology and other resources to facilitate remote working. Overall, the study provides valuable information for organizations seeking to implement remote working and for policymakers looking to improve working conditions and productivity in the public sector.

The paper proceeds as follows: Section 2 reviews the related literature on remote working. Section 3 demonstrates the theoretical framework and develops the study hypotheses. Section 4 illustrates the research methodology. The data analysis techniques and results are presented in Section 5. Section 6 concludes the paper.

## 2. Literature Review

Recently, considerable literature has grown around accountants shifting to remote work during the COVID-19 pandemic. However, remote working has been defined over time in different contexts. Remote working was initially defined by Nilles [13] as "a subset of teleworking" to reduce traffic. In addition, Vyas and Butakhieo [14] defined remote working within the COVID-19 context as "an alternative working to minimize the risk of COVID-19 infection", while Gajendran and Harrison [15] defined the concept as "an alternative work arrangement in which employees perform tasks elsewhere that are normally done in a primary or central workplace, for at least some portion of their

work schedule, using electronic media to interact with others inside and outside the organization". Moreover, many authors point to remote working using different terms, such as work from home, telework, e-working, and online jobs. These terms are similar in their use of technology to allow employees to work in flexible and distanced workplaces to execute work duties [14,16].

Further, authors have asserted that adopting remote jobs is mainly associated with a positive effect on the company's productivity and flexibility, as well as a reduction of conflicts in work–life [17]. In contrast, some authors linked the low level of well-being, reliable information, and performance and high pressure on employees who work remotely [18]. However, Dingel and Neiman [19] reported that 37 percent of jobs could be completed at home during the pandemic of COVID-19, such as accountants, managers, and other scientific services. However, as the acceptance of remote working has grown worldwide, authors have examined the advantages and disadvantages of remote working adoption. Accordingly, in the context of governmental institutions, authors have shown that remote working has developed a policy urgency for governmental institutions to handle remote working effectively during the pandemic. Many authors have reported the usefulness of remote working adopted by some countries' governmental authorities, such as Hong Kong and other Asian countries [14].

Crises tend to encourage embedded business-related behaviours to adapt to the current situation, such as technology adoption for remote jobs [20]. Scholars have used different ways to interpret the transfer to remote working. For example, some authors have emphasized the cost-effectiveness and suitability of firms' productivity in adopting remote working in different institutions [21].

Furthermore, according to Leoni, Lai [1], much literature has showed the accountant's role in supporting companies and governments in their response to the pandemic. Mitchell, Nørreklit [22] explored three countries, namely Germany, Italy, and the UK, finding that lockdown and online work conversion behaviours are specifically accountants' responsibility. In addition, Delfino and van der Kolk [4] investigated the broad scope of remote working for various professions in firms. They used an interview method with over 15 employees in different departments, investigating their responses to the use of new techniques. The findings show that technology, digital procedures, and controls have negatively impacted management controls and systems.

On the other hand, some scholars have been more specific and taken accounting as an examination field; for example [23]. Some authors used models and conceptual frameworks to examine accountants' acceptance of technology while remotely working during the coronavirus crisis [24,25]. Handoko, Muljo [25] equipped factors from the Technology Acceptance Model (TAM), Technology–Organization–Environment model (TOE), and Unified Theory of Acceptance and Use of Technology (UTAUT) and integrated them to explore the main effects of remote working on accountants and auditors. Many scholars have used a structured theoretical model to understand the impacts of technological conversion during the pandemic, such as in the educational field Alshurafat, Al Shbail [26] and the healthcare sector [27].

Generally speaking, the technology acceptance phenomena has been under examination for more than 3 decades [28]. Literature review papers have confirmed the high potentiality of research in this area and the plethora of research gaps in technology acceptance topics [28–30]. Most scholars in this area asserted that the reasons why technology acceptance remains an area of scientific examination are twofold [28,31,32]. First, the theoretical model regarding technology acceptance could be easily integrated or expanded, allowing for exploring a technology's acceptance from different angles [28,29,33–37]. Second, technology is rapidly developing, allowing for new technology acceptance examinations [38,39].

## 3. Theoretical Framework and Research Hypotheses

*3.1. Research Model*

Several research models seek to justify the acceptance of technology. Therefore, as shown in Figure 1, this study provides an integrated model from a set of theories, including TAM, TOE, SCT, and TRA.

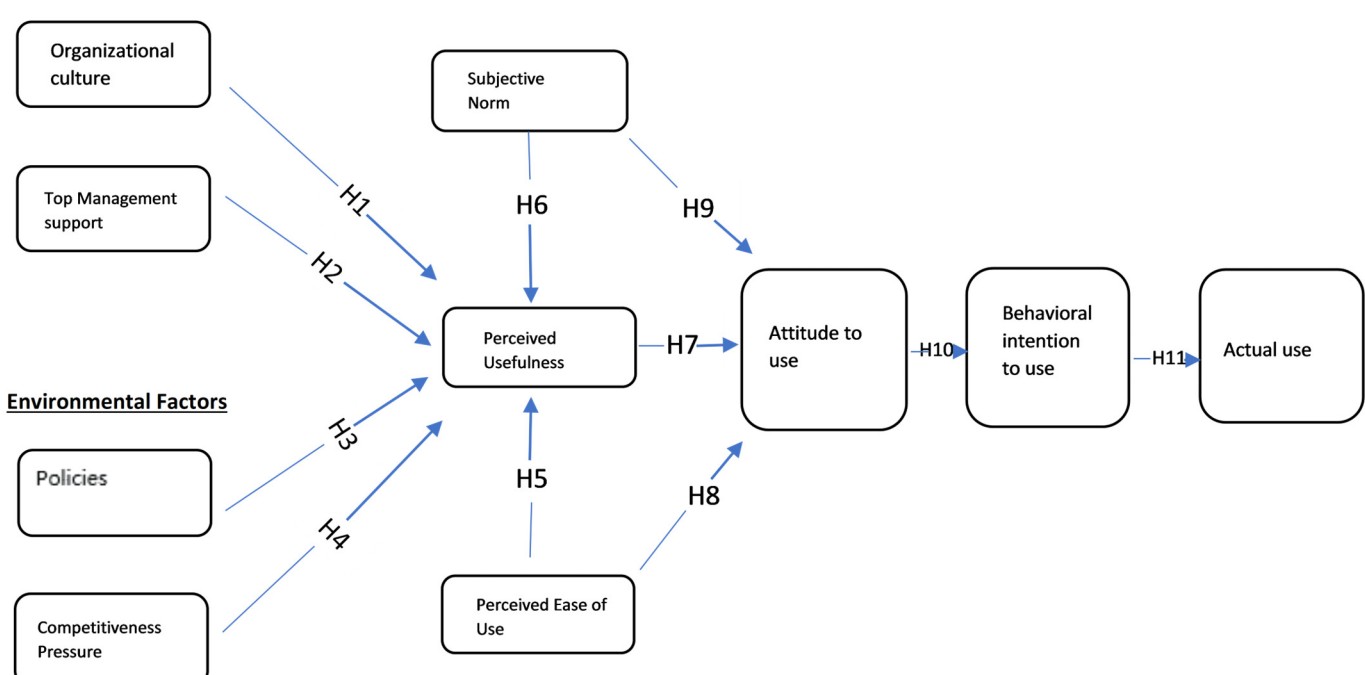

**Figure 1.** Theoretical research model.

TAM, TOE, SCT, and TRA are commonly used theoretical frameworks in the field of information systems and technology [10,11,40–45]. These frameworks have been extensively studied and are well-established in the literature, which is why researchers have chosen to analyze these dimensions in their research. Using these theoretical frameworks, researchers can examine the various factors that influence the adoption and use of technology comprehensively and systematically [38,46]. This allows researchers to better understand the underlying mechanisms that influence user behaviour and to develop a more nuanced understanding of the complex relationships between the different variables.

Furthermore, using established theoretical frameworks provides a solid foundation for research and helps ensure that findings are grounded in existing knowledge and are well-supported by prior research. This can increase the validity and reliability of results and make it easier to compare findings with previous studies in the field.

The first model, the TAM, illustrates how users accept a technology [4]. It consists of two main factors affecting the intention to use a technology: perceived ease of use and perceived usefulness [47]. The second model is the TOE model. This framework explains factors that influence decisions to adopt a technology, such as organizational culture, top management, policies, and competitive pressures [48]. The third model is the SCT; this framework can be seen as a social framework that facilitates specific acts by individuals that benefit accountants and organizations [49]. The fourth model is the TRA, which includes the attitudes factor, which is ideas about a specific object or action that can be translated into a desire or intention to act [50]. The TRA also includes two other factors: intention to use and user behaviour. Many previous studies have considered these factors when examining technology acceptance behaviour.

### 3.2. Organizational Culture and the Perceived Usefulness

Hofstede, Hofstede [51] defined organizational culture as shared programming for the mind, which distinguishes the members of one organization from other organizations. Moreover, according to O'reilly and Chatman [52], organizational culture is defined as "shared norms and values that set expectations about appropriate attitudes and behaviour for group members". Authors in technology acceptance research have focused more on the relationships between the organizational culture and other factors, such as perceived usefulness or ease of use, since neglecting cultural differences can prevent the adoption of information technology and increase failure [53].

Perceived usefulness is one of the significant factors impacted by organizational culture. Perceived usefulness is defined by [54] as "the prospective user's subjective probability that using a specific application system will increase his or her job performance within an organizational context".

Furthermore, Hwang, Al-Arabiat [55] found that organizational culture influences the technological factors of the parties involved in the implementation process and how they use them. In a remote working context, it was found that different organizational cultures regularly possess different values, obligations, and expectations that directly or indirectly influence the level of benefit accrued from remote working in firms [56]. Therefore, organizational culture is essential because it influences individual behaviour and helps shape the organizational climate, which can assist the organization in achieving its objectives [57]. The lack of an excellent organizational culture motivated by technology's perceived usefulness impedes the successful adoption of remote working [58]. Therefore, the following hypothesis is stated:

**H1.** *Organizational culture positively impacts the perceived usefulness of remote working for accountants within Jordanian governmental organizations.*

### 3.3. Top Management Support and the Perceived Usefulness

Top management support refers to "decision-makers who influence innovation adoption" [59]. Top management means perceptions and actions regarding the utility of technological innovation in creating value for organizations [54]. Scholars found a significant positive relationship between top management support and the adoption of remote working [60]. Wang, Liu [20] stated that top management support is one of the critical predictors for adopting remote working on an organizational level.

Top management support helps create positive environments, ensures acceptable sources of innovation adoption, and even manages the diffusion procedures for acceptance of innovations among organization members, determining the allocation of resources needed to support innovation implementation. Adopting remote working in an organization is a strategic decision [61]. Conversely, where managers do not support adopting remote working, remote working work fails to produce positive results [62]. Several empirical studies showed a positive relationship between top management support and perceived usefulness [63,64]. Therefore, the following hypothesis is stated:

**H2.** *Top management support positively impacts the perceived usefulness of remote working for accountants within Jordanian governmental organizations.*

### 3.4. Policies and Perceived Usefulness

Governmental laws and regulations show clear guidelines for companies regarding information technology [54]. According to Deephouse [65], governmental laws and legal frameworks shape an organization's technology adoption. The literature has found that perceived governmental pressure is critical in technology adoption. The decision to adopt an information system is impacted by governmental pressure [66]. Moreover, studies have shown that governmental laws can encourage or discourage innovation [67], which the government can support through financial incentives, pilot projects, and tax breaks to stimulate technological innovation in firms [68].

Based on Alshaikh, Maasher [69], governmental laws can mandate allocating resources for compliance with specific systems or technology adoption. Evidence supports that governmental laws influence an organization's ability to provide or adopt technology [70]. Moreover, studies have shown a positive relationship between governmental regulations encouraging innovation adoption in different contexts, such as blockchain adoption [71]. Accordingly, Basole, Seuss [72] claimed that government laws are a critical factor, playing a massive role in the adoption of technology within an organization. In addition, Tutusaus, Schwartz [73] stated that government laws had been included in prior studies as prominent and influential factors influencing the adoption of innovations. Therefore, the following hypothesis is stated:

**H3.** *Governmental policies regarding remote working positively impact the perceived usefulness of remote working for accountants within Jordanian governmental organizations.*

*3.5. Competitive Pressure and Perceived Usefulness*

Scholars have defined competitive pressure as a firm's response to the level of uncertainty that competitors in an industry can directly impose on an organization [74]. Jeyaraj, Rottman [75] identified competitive pressures as one of the most significant predictors in terms of organizations adopting new and advanced technologies. Scholars have recognized competitive pressure as a diffusion driver for technology presence in firms [76,77]. Further, Ramdani, Kawalek [78] argued that it is evident that technological innovation affects competition. In addition, findings have suggested that intense competition is a critical determinant in the adopting of information technology systems for a firm's operations [79].

In the context of remote working, Soroui [80] found that shifting a firm's logic to remote working is affected positively by the competitive pressure to adapt quickly, therefore ensuring they do not lose out. Other scholars found that remote working utilization can trade off between the pressure that stems from competitors and obtaining control or coordination [81]. Furthermore, prior studies support that pressures from high competition lead to imitating competitors' adoption decisions [82], such as switching to remote working. Accordingly, Ramdani, Kawalek [78] asserted that adopting new technologies can become a strategic requirement to compete and be more sustainable in the marketplace. Additionally, pressure from competitors positively impacts a firm's technology diffusion [76,83]. Therefore, the following hypothesis is stated:

**H4.** *Competitive pressure positively impacts the perceived usefulness of remote working for accountants within Jordanian governmental organizations.*

*3.6. Subjective Norms, Perceived Usefulness, Perceived Ease of Use, Attitude toward Use, and Behavioural Intention to Use*

The subjective norms construct is derived from the theory of reasoned actions [84]. Subjective norms refer to the degree to which an individual perceives that those significant or influential to them believe they should engage in a certain behaviour [84]. According to Teo [85], the subjective norm is "a person's perception that most people who are important to him or her think he or she should or should not perform the behaviour in question". In addition, subjective norms are forms of social pressure from others that significantly influence someone's behaviour [86]. Based on Awa, Ojiabo [87], many psychological studies theorize that subjective norms are an essential determinant of intention, and especially epitomize the perception of others in terms of behaviour.

Attitude is defined by Kashif, Zarkada [88] as "judgment of outcomes concerning a particular behaviour", while subjective norms take a holistic perspective and consider the role of family, friends, and peers in approving or disapproving of certain behaviours. Perceived ease of use refers to "the degree to which a person believes that using a particular system would be free of effort". In contrast, perceived usefulness refers to "the degree to which a person believes that using a particular system would enhance his or her job performance" [89]. In some previous studies, the attitude construct has been removed

to provide a direct non-mediated relationship between perceived usefulness, perceived ease of use and behavioural intention [90]. However, this study intends to stick with the original construction of the TAM. Therefore, attitude is included in this study's integrated theoretical model.

Previous scholars found that attitude and subjective norms influence the intention to perform a specific behaviour [91]. In addition, in terms of subjective norms attitudes [92], when a person perceives an action as positive, it is more likely to impact their intention to engage in that behaviour. As a result, two vital signs—attitude toward the behaviour and subjective norms—affect the intention to use [93]. At the individual level, attitude, subjective norms, and perceived usefulness help predict working professionals' behaviour [88]. Davis and Fred [89] discovered strong relationships between attitudes, ease of use, and perceived usefulness. Attitudes toward adopting remote working are affected by perceived usefulness and perceived ease of use [26].

The perceived usefulness and perceived ease of use significantly influence attitudes toward use [85]. Attitude is a crucial determinant of behavioural intention and positively influences accountants' intention to adopt remote working [92]. Perceived benefit and perceived usefulness directly impact accountants' attitudes toward the intention to adopt remote working [94]. Furthermore, there is a direct impact between attitudes toward adopting remote working and behavioural intentions to adopt remote working [95]. Therefore, the following hypotheses are stated:

**H5.** *Perceived ease of use positively impacts the perceived usefulness of remote working for accountants within Jordanian governmental organizations.*

**H6.** *Subjective norms positively impact the perceived usefulness of remote working for accountants within Jordanian governmental organizations.*

**H7.** *Perceived usefulness positively impacts attitudes toward using remote working for accountants within Jordanian governmental organizations.*

**H8.** *Perceived ease of use positively impacts attitudes toward using remote working for accountants within Jordanian governmental organizations.*

**H9.** *Subjective norms positively impact attitudes toward using remote working for accountants within Jordanian governmental organizations.*

**H10.** *Attitudes toward use positively impact behavioural intention to use remote working for accountants within Jordanian governmental organizations.*

*3.7. Intention to Use and Actual Use*

Intention to adopt is a psychological state that occurs just before a person adopts new technology [96]. Behavioural intention is used to define an intention to use the service, which is defined as "the strength of one's intention to perform a specified behaviour" [84]. The importance of behavioural intention as a predictor of individual behaviour is well documented in accounting literature [97]. This relationship is also emphasized in the TRA and TAM theoretical models [89]. In most studies of remote working adoption, it is empirically evident that actual use is mainly impacted by the behavioural intention to use [26]. Teo [85] asserted that intention is a determinant of denoting the factors that influence desired behaviour (e.g., the use of remote working).

Based on previous scholars, technology adoption shows the importance of intention in predicting actual use [98]. Thus, there is a positive relationship between a high level of intention to adopt technology and a high level of actual adoption of the technology [26,99]. Therefore, the following hypothesis is stated:

**H11.** *Behavioural intention to use positively impacts the actual use of remote working for accountants within Jordanian governmental organizations.*

## 4. Methodology

### 4.1. Questionnaire Design and Data Collection

The questionnaire was divided into two sections: Section A provides the participants' demographic information, and Section B covers the model's various construct measurements. Furthermore, the questionnaire was written in English and translated into Arabic, Jordan's native tongue. A pilot investigation of 15 samples confirmed the phrasing, structure, content, arrangement, layout, simplicity, and clarity of the items inside the questionnaire. The pilot study results were used to improve the effectiveness of the final questionnaire.

To reiterate, the primary data collecting instrument was a questionnaire, of which copies were generated and gathered to collect the data. The study respondents are accountants working in Jordanian government organizations who are well-educated and informed about the research topic, and all of them are native to the language of the survey. The authors used Microsoft office forms, believing this was the best approach to ensuring a high response rate, as Cobanoglu and Cobanoglu [100] suggested. An online survey is advantageous since it is less expensive, offers quick results, and covers a large geographic area [101]. It has also been used widely in earlier studies [102]. When the population is unknown and obtaining replies from the complete sampling frame is challenging, it is seen as more acceptable [103].

Additionally, the non-probability sampling technique is better suited for theoretical generalization [104]. As a result, the respondents for this study were chosen using a purposive sample technique. One hundred and seventy copies were collected from the distributed copies, with fifteen surveys being rejected due to incomplete information or unanswered questions. As a result, the whole statistical analysis includes 155 questions. The number of returned surveys is considered adequate [105].

The sample size of 155 respondents can be considered adequate for obtaining valid research results in a PLS-SEM (Partial Least Squares Structural Equation Modelling) analysis as long as certain conditions are met [106]. However, it is worth noting that there is no definitive rule for determining an optimal sample size, and the appropriate sample size depends on several factors, including the research question, the complexity of the data, the number of variables being studied, and the desired level of precision [107].

PLS-SEM relies on latent variables (latent factors) to explain the relationships among the observed variables rather than requiring large sample sizes to estimate complex covariances or parameters [108].

For a sample size of 155, the number of latent variables and the number of indicators per latent variable (i.e., the number of observed variables that measure each latent variable) are important factors to consider when determining the validity of the results. In general, having at least 10–15 observations per latent variable is recommended, which would mean that, with 155 respondents, the number of indicators per latent variable should be kept to a reasonable number (e.g., less than 11). In this study, 10 latent variables are examined. Therefore, 155 respondents is considered adequate for obtaining valid research results.

### 4.2. Research Instruments

The research instruments include two critical parts: the first complied with the nominal scales for basic information collection, and the second complied with the 5-point Likert scales. The first part includes basic information questions to collect accountants' characteristics, such as qualifications, gender, and age. In addition, the second part consists of 11 constructs that measure the determinants of actual use by the SCT, TAM, TOE, and TRA theories [84,109].

This paper used existing scales to measure constructs related to organizational culture, top management, policies, competitive pressure, subjective norms, perceived usefulness, perceived ease of use, attitude toward use, behavioural intention to use, and actual use.

For the organizational culture construct, three items were adapted from the scales developed by Pillai, Sivathanu [110] and Gangwar [111]. Similarly, three items used to mea-

sure the top management construct were adapted from Abed [42] and Gangwar [111], and three items used to measure the policies construct were adapted from Pillai, Sivathanu [110]. The construct of competitive pressure was measured using three items adapted from Sun, Hall [61].

The constructs of subjective norms, perceived usefulness, perceived ease of use, attitude toward use, and behavioural intention to use were each measured using three items adapted from the scales developed by Davis [89], Alshurafat, Al Shbail [26] and Daragmeh, Sági [9]. Four items were used to measure subjective norms while two items were used to measure actual use, adapted from the scales developed by Davis [89], Alshurafat, Al Shbail [26], and Daragmeh, Sági [9]. The items were adapted from this previous work to ensure that they were relevant and appropriate for the current study's research questions and context, with changes made to improve clarity and fit for the study's purposes while maintaining their original intent and meaning.

### 4.3. Data Analysis Technique

Data analysis was conducted using PLS-SEM in Smart-PLS, Version 3.0, for testing the formulated study hypotheses. Descriptive analysis also has been used to describe the data. The analysis shows the descriptive statistical information about the frequency of occurrence, the average score or central tendency, known as the mean, and the range of variability, known as a standard deviation—all combined with a graphical representation of the data. All valid responses are subjected to the partial least square structural method (PLS-SEM) analysis approach as per the recommendation of [107]. This approach is used to measure the behavioural and social sciences. The tool can model latent variables, rectify measurement errors, and evaluate all model parameters simultaneously. In addition, Smart-PLS 3.0 was used for model measurement to determine the relevant structure equation following the guidelines stated by Hair Jr., Hult [107].

Many researchers use the PLS-SEM method as it allows them to estimate complex models with many constructs, indicator variables, and structural routes without applying distributional assumptions to the data. PLS-SEM is a causal-predictive approach to SEM that focuses on prediction when estimating statistical models with structures that aim to provide causal explanations [107]. Moreover, PLS-SEM eliminates the apparent gap between confirmatory and predictive research, following Hair Jr., Hult [107]. The PLS-SEM method was used in the current study. This is an appropriate strategy for several reasons: (1) it focuses on predicting endogenous variables, in accordance with Aburumman, Omar [106]; (2) the study research model is a highly complicated one; and (3) predictive modelling was stressed in earlier studies that showed PLS-SEM to be appropriate for studies on technology acceptance and adoption [44].

Overall, the above justifies using Smart-PLS path modelling to confirm the measurement and structural models—the measurement model explains the constructs, reliability, and validity. In contrast, structural analysis simultaneously provides the bivariate correlation analysis and regression analysis to determine the relationships of the study constructs. The PLS algorithm and bootstrapping method were used to determine factors affecting remote working adoption during the COVID-19 pandemic among accountants in Jordanian governmental organizations.

### 4.4. Common Method Bias (CMB) Test

The issue of common method bias (CMB) was expected, given that this study used a self-report survey in which the independent and dependent variables came from the same respondents. As a result, a full collinearity test was conducted for this study, as suggested by Kock and Lynn [112], who stated that if the VIF is greater than 3.3, then CMB and collinearity are present in the model. This study's obtained VIF from the full collinearity test was less than 3.3. Therefore, it can be assumed that CMB is absent from the model. As a result, the data would not cause survey measures to be misinterpreted.

## 5. Results

### 5.1. Demographics

As shown in Table 1, there are relative proportions of males and females in Jordanian government organizations. Nearly half of the respondents (42.2%) were between 31 and 40 years old. Most respondents (82%) possessed a qualification and an accounting certification. Most respondents (67%) worked as an accountant, while one-third (31%) were highly experienced, with 6 to 10 years of working experience.

**Table 1.** Respondents' demographic information (n = 155).

| Gender | Frequency | Percent |
|---|---|---|
| Male | 83 | 54% |
| Female | 72 | 46% |
| 30 or below | 41 | 29% |
| 31–40 | 50 | 42.2% |
| 41–50 | 35 | 20.4% |
| 51-Years or Above | 29 | 8.4% |
| Diploma | 11 | 7% |
| Bachelor | 95 | 61% |
| Master | 45 | 29% |
| PhD | 4 | 3% |
| Less than 5 years | 43 | 28% |
| 6–10 years | 48 | 31% |
| 11–15 years | 41 | 26% |
| 16 and above | 23 | 15% |
| Accountant | 104 | 67% |
| Account Manager | 5 | 3% |
| Financial Manager | 7 | 5% |
| Other | 39 | 25% |
| CIA | 11 | 7% |
| CPA | 15 | 10% |
| CISA | 7 | 5% |
| CRISC | 2 | 1% |
| JCPA | 38 | 24% |
| Other | 54 | 35% |
| None | 28 | 18% |
| **Total** | **155** | **100%** |

### 5.2. Evaluation of PLS-SEM Results

The results of the factor analysis are highlighted in this section. To conclude, all the items were adapted from previous studies. The construct's measures are also assessed for their reliability and validity. In the context of factor analysis, the outer model infers the unidimensionality of the research variables. After confirming the created measure's reliability and validity, the structural models and the relationships between the latent variables were examined.

Following the data checking and screening described previously, Hair Jr., Hult [107] noted that the next stage was to analyze the outer and inner models. This study applied the PLS-SEM to the outside (measurement model) and interior models (structural model) to achieve this aim. To analyze the direct findings of this investigation, PLS-SEM was used. Furthermore, Smart-PLS 3.0 was used to determine the causal relationships between the constructs in these theoretical models [113].

5.2.1. Measurement Model Assessment

The validity and reliability of the measurement model are evaluated by assessing (1) internal consistency reliability; (2) indicator reliability; (3) convergent validity; and

(4) discriminant validity. The following sections present the results of all analyses to evaluate the validity and reliability of the measurement model.

A measurement model is said to have satisfactory internal consistency reliability when each construct's composite reliability (CR) exceeds the threshold value of 0.7 [107]. Table 2 shows that the CR of each construct for this dissertation ranges from 0.82 to 0.909. These results indicate that the constructs' items pose satisfactory internal consistency reliability.

**Table 2.** Measurement scales, reliability, convergent validity, and VIF results.

| Construct | Items | Loadings | Cronbach's Alpha | CR | AVE | VIF |
|---|---|---|---|---|---|---|
| **Organizational culture** | OC.1 | 0.778 | 0.770 | 0.820 | 0.603 | 1.327 |
| | OC.2 | 0.762 | | | | 1.274 |
| | OC.3 | 0.789 | | | | 1.313 |
| **Top management support** | MS.1 | 0.768 | 0.727 | 0.845 | 0.646 | 1.460 |
| | MS.2 | 0.801 | | | | 1.373 |
| | MS.3 | 0.807 | | | | 1.482 |
| **Policies** | PO.1 | 0.790 | 0.762 | 0.861 | 0.675 | 1.558 |
| | PO.2 | 0.849 | | | | 1.473 |
| | PO.3 | 0.823 | | | | 1.633 |
| **Competitive pressure** | CP.1 | 0.741 | 0.752 | 0.857 | 0.668 | 1.308 |
| | CP.2 | 0.875 | | | | 1.737 |
| | CP.3 | 0.831 | | | | 1.793 |
| **Subjective norms** | SN.1 | 0.848 | 0.822 | 0.882 | 0.651 | 2.188 |
| | SN.2 | 0.756 | | | | 1.705 |
| | SN.3 | 0.821 | | | | 1.836 |
| | SN.4 | 0.801 | | | | 2.002 |
| **Perceived ease of use** | PE.1 | 0.842 | 0.766 | 0.864 | 0.680 | 1.553 |
| | PE.2 | 0.805 | | | | 1.591 |
| | PE.3 | 0.826 | | | | 1.532 |
| **Perceived Usefulness** | PU.1 | 0.808 | 0.790 | 0.877 | 0.704 | 1.522 |
| | PU.2 | 0.863 | | | | 1.857 |
| | PU.3 | 0.845 | | | | 1.720 |
| **Attitudes toward use** | AT.1 | 0.708 | 0.705 | 0.835 | 0.630 | 1.264 |
| | AT.2 | 0.793 | | | | 1.454 |
| | AT.3 | 0.872 | | | | 1.587 |
| **Behavioural intention to use** | BI.1 | 0.831 | 0.777 | 0.870 | 0.690 | 1.549 |
| | BI.2 | 0.830 | | | | 1.721 |
| | BI.3 | 0.832 | | | | 1.575 |
| **Actual Use** | AU.1 | 0.936 | 0.803 | 0.909 | 0.833 | 1.819 |
| | AU.2 | 0.889 | | | | 1.819 |

The indicator reliability of the measurement model is measured by examining the loadings of the items. A measurement model has satisfactory indicator reliability when each item's loading estimate is higher than 0.7 [107]. Based on the analysis, all items in the measurement model exhibited loadings exceeding 0.7, ranging from 0.708 to 0.936. All items are significant at the level of 0.001. Table 2 shows the loading for each item. Thus, all items used for this study demonstrate satisfactory indicator reliability.

This paper assesses the measurement model's convergent validity by examining its average variance extracted (AVE) value. Convergent validity is adequate when constructs have an average variance extracted (AVE) value close to 0.5 or higher [107]. Table 2 shows that all constructs have an AVE ranging from 0.603 to 0.833. suggesting the measurement model exhibits adequate convergent validity.

The measures' discriminant validity comprises the degree to which items differ among constructs or measure distinct concepts. In other words, as indicated by Hair Jr., Hult [107],

the measures of constructs that are theoretically not linked to each other have no linkage to each other. Hair Jr., Hult [107] mentioned the Fornell–Larcker criterion as the most conventional approach to assessing discriminant validity, while Henseler, Ringle [114] mentioned the widespread usage of the HTMT for the same purpose.

The Fornell–Larcker criterion: This method has traditionally been used to assess discriminate validity. When the square root value of the AVE of each construct is greater than the construct's highest correlation with any other latent construct, discriminant validity is confirmed [114]. To evaluate the discriminant validity, this study computes the square root of the AVE for each construct with the correlations presented in the correlation matrix. The results of the Fornell–Larcker criterion assessment with the square root of the constructs are shown in Table 3. Based on the results, it can be said that the discriminant validity of the construct is established based on the guidelines [114].

**Table 3.** Discriminant validity.

| Fornell–Larcker Criterion | 1 | 2 | 3 | 4 | 5 | 6 | 7 | 8 | 9 | 10 |
|---|---|---|---|---|---|---|---|---|---|---|
| Organizational culture | 0.794 | | | | | | | | | |
| Top management | 0.177 | 0.913 | | | | | | | | |
| Policies | 0.435 | 0.187 | 0.831 | | | | | | | |
| Competitive pressure | 0.143 | 0.041 | 0.223 | 0.817 | | | | | | |
| Subjective norms | 0.207 | 0.176 | 0.327 | 0.269 | 0.804 | | | | | |
| Perceived ease of use | 0.279 | 0.138 | 0.441 | 0.308 | 0.615 | 0.776 | | | | |
| perceived usefulness | 0.445 | 0.159 | 0.521 | 0.351 | 0.325 | 0.494 | 0.825 | | | |
| Attitudes toward use | 0.165 | 0.251 | 0.190 | 0.194 | 0.284 | 0.328 | 0.176 | 0.821 | | |
| Behavioural intention to use | 0.484 | 0.184 | 0.537 | 0.290 | 0.590 | 0.648 | 0.551 | 0.380 | 0.839 | |
| Actual use | 0.411 | 0.102 | 0.277 | −0.002 | 0.200 | 0.189 | 0.284 | 0.100 | 0.365 | 0.807 |

Heterotrait–monotrait ratio (HTMT): HTMT, introduced by Henseler, Ringle [114], is a novel method for investigating the discriminant validity of latent variables. In the context of PLS-SEM, the HTMT is used to assess the discriminant validity of the constructs. The HTMT method was used as a more stringent criterion during this stage than the other two traditional approaches. Smart-PLS software is used to calculate the HTMT criterion. Smart-PLS was used to compute the HTMT correlations in this study. Table 4 contains the HTMT criterion results, which show that the HTMT criterion is less than 0.791 for each latent variable. Thus, discriminant validity is established using the HTMT method [114].

**Table 4.** Heterotrait–monotrait ratio.

| HTMT Criterion | 1 | 2 | 3 | 4 | 5 | 6 | 7 | 8 | 9 | 10 |
|---|---|---|---|---|---|---|---|---|---|---|
| Organizational culture | | | | | | | | | | |
| Top management | 0.236 | | | | | | | | | |
| Policies | 0.574 | 0.236 | | | | | | | | |
| Competitive pressure | 0.224 | 0.113 | 0.277 | | | | | | | |
| Subjective norms | 0.290 | 0.237 | 0.432 | 0.347 | | | | | | |
| Perceived ease of use | 0.402 | 0.182 | 0.605 | 0.422 | 0.824 | | | | | |
| perceived usefulness | 0.597 | 0.205 | 0.669 | 0.447 | 0.438 | 0.686 | | | | |
| Attitudes toward use | 0.227 | 0.306 | 0.244 | 0.250 | 0.386 | 0.460 | 0.224 | | | |
| Behavioural intention to use | 0.642 | 0.233 | 0.684 | 0.367 | 0.771 | 0.791 | 0.702 | 0.480 | | |
| Actual use | 0.527 | 0.135 | 0.359 | 0.111 | 0.243 | 0.243 | 0.362 | 0.122 | 0.439 | |

As a result, analysis of the outer model confirms that the survey items measure the constructs they were designed to measure. In other words, the items are trustworthy and reliable.

### 5.2.2. Structural Model Assessment

To examine the inner model, the first step's prerequisites must be met (measurement model assessment). The study's final portion covered the necessary criteria for evaluating the measurement model's quality. The inner model assessment requirements for this study

are explained in this section. The model's capacity to forecast and the link between latent variables are addressed in detail.

This study applied these key factors to assess the inner model. They are as follows: endogenous constructs' coefficient of determination ($R^2$); effect size and predictive relevance or cross-validated redundancy ($Q^2$); model fit; and path coefficients and their significance (standard errors, significance levels, t-values, and *p* values).

In the structural model evaluation, the endogenous latent variables' coefficient of determination (R2) is one of the most used metrics [115]. Chin [116] criterion posits that the $R^2$ value equivalent to or higher than 0.67 is substantial, while 0.33 is moderate and 0.19 is weak. Based on these criteria, the $R^2$ of the endogenous variables perceived usefulness, attitude toward use, behavioural intention to use, and actual use are 0.595, 0.333, 0.189, and 0.035, correspondingly, and are referred to in Figure 2. These values are deemed high, reflecting the established model's sufficiency.

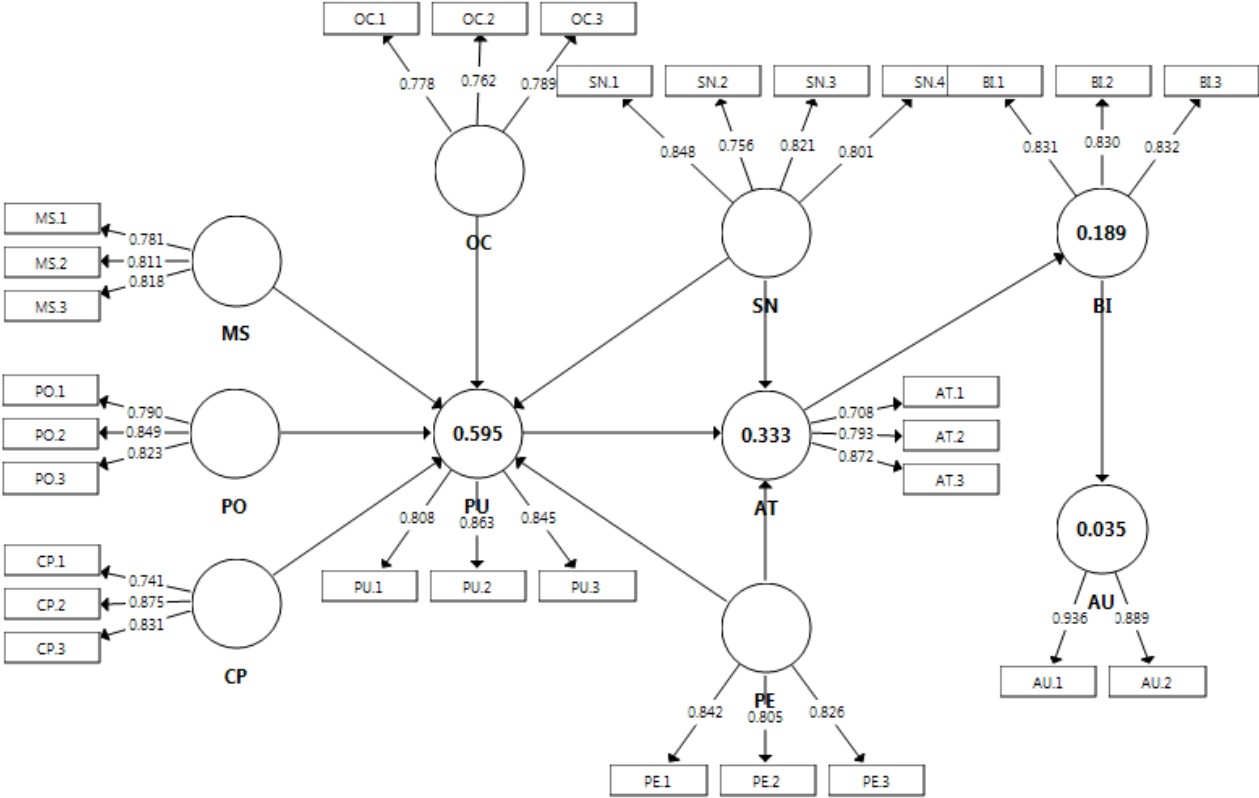

**Figure 2.** Item loadings and R² values.

Stone–Geisser's Q-test is a model fit indicator. It determines the model's ability to predict parameter estimation. The Q-test value is obtainable by PLS software using the blindfolding routine. For this purpose, the number of the omission distance is fixed at seven during the blindfolding procedure. Further, the obtained cross-validated redundancy values for each endogenous latent variable are greater than zero, as Tenenhaus, Vinzi [117] recommended; specifically, 0.579 for perceived usefulness, 0.320 for 0.184 attitude toward use, 0.184 for behavioural intention to use, and 0.029 for actual use (refer to Table 5). The estimated results show that the structural model could predict the related estimations due to the good reconstruction of values [114].

**Table 5.** $R^2$, $R^2$ Adjusted, and $Q^2$.

| Variable | Variable Type | R-Square | R-Square Adjusted | Q-Squared |
|---|---|---|---|---|
| **Perceived usefulness** | Endogenous | 0.595 | 0.579 | 0.375 |
| **Attitude toward use** | Endogenous | 0.333 | 0.320 | 0.196 |
| **Behavioural intention to use** | Endogenous | 0.189 | 0.184 | 0.125 |
| **Actual use** | Endogenous | 0.035 | 0.029 | 0.013 |

After determining the goodness of the outer mode, the hypothesized associations between the constructs were tested. The hypothesized model was tested using the PLS algorithm and Smart-PLS (v.3.0). The path coefficients were then generated, as shown in Figure 3.

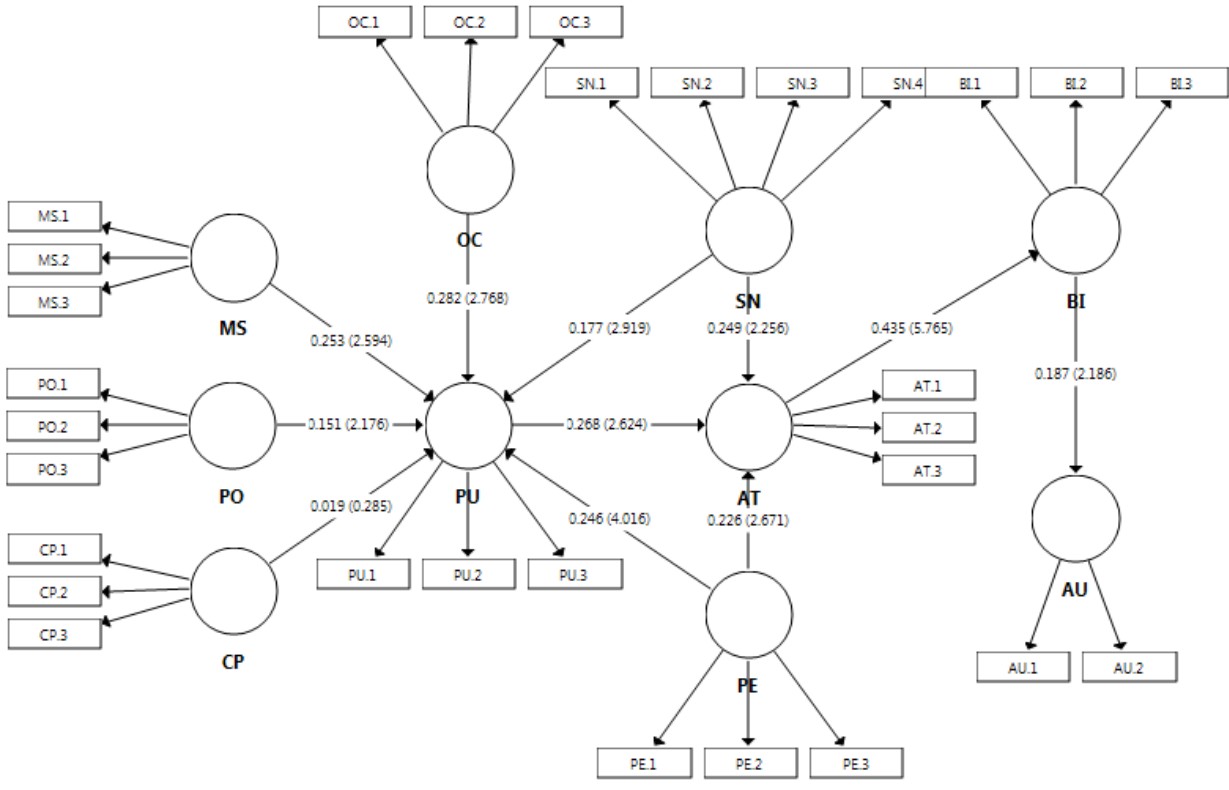

**Figure 3.** PLS bootstrapping (t-values) for the study model.

The application of the bootstrapping method in assessing the path coefficients includes a bootstrap sample of 5000 minimum. At the same time, the number of cases must tally with the total observations in the initial sample [107]. Likewise, for a one-tailed test, the critical t-values are 1.65 (with a significance level of 5%). Correspondingly, to generate standard errors and obtain t-statistics, 5000 re-sampling with a replacement number from the bootstrap cases equivalent to the original number of the sample (155) was set. The path coefficient and the bootstrapping results are presented in Figure 3 and Table 6, where the hypothesized relationships (as shown below) were tested.

This study's structural model has 11 direct hypotheses addressing 10 variables. They were tested using PLS-SEM, with six exogenous constructs tested against four endogenous constructs (perceived usefulness, attitude toward use, behavioural intention to use, and actual use) in stage one. The same four endogenous constructs were examined in stage two against perceived usefulness. In stage three, attitude toward use was tested against behavioural intention to use, and in stage three, behavioural intention to use was tested against actual use. Figure 3 indicates that the fit measure in the final structural model has a good fit with three paths at the 5% significance level ($p < 0.05$).

**Table 6.** Hypotheses verification.

| No | Hypotheses | Path Coefficient | t Statistics | *p*-Values | Decision |
|---|---|---|---|---|---|
| H1 | Organizational culture → Perceived usefulness | 0.282 | 2.768 | 0.006 * | *Supported* |
| H2 | Management support → Perceived usefulness | 0.253 | 2.594 | 0.010 * | *Supported* |
| H3 | Policies → Perceived usefulness | 0.151 | 2.176 | 0.030 * | *Supported* |
| H4 | Competitive pressure → Perceived usefulness | 0.019 | 0.285 | 0.776 | *Not Supported* |
| H5 | Perceived ease of use → Perceived usefulness | 0.246 | 4.016 | 0.000 * | *Supported* |
| H6 | Subjective norms → Perceived usefulness | 0.177 | 2.919 | 0.004 * | *Supported* |
| H7 | Perceived usefulness → Attitude toward use | 0.268 | 2.624 | 0.009 * | *Supported* |
| H8 | Perceived ease of use → Attitude toward use | 0.226 | 2.671 | 0.008 * | *Supported* |
| H9 | Subjective norms → Attitude toward use | 0.249 | 2.256 | 0.024 * | *Supported* |
| H10 | Attitude toward use → Behavioural intention to use | 0.435 | 5.765 | 0.000 * | *Supported* |
| H11 | Behavioural intention to use → Actual use | 0.187 | 2.186 | 0.029 * | *Supported* |

*: $p < 0.05$.

Table 6 shows that hypothesis 1 is supported ($\beta$ = 0.282, t = 2.768, $p$ = 0.006), indicating that organizational culture positively influences perceived usefulness. This means that the organizational culture in Jordanian governmental organizations leads to high levels of perceived usefulness. Moreover, management support was found to relate significantly to perceived usefulness, with the structural path coefficient between top management support and perceived usefulness, addressed in hypothesis 2, being 0.253. The positive sign of the coefficient shows the positive effect of top management support on perceived usefulness. This means that hypothesis 2 is accepted.

Moving on to hypotheses 3 and 4, the effect of policies on perceived usefulness and competitive pressure on perceived usefulness was addressed. While the results supported hypothesis 3, hypothesis 4 is not supported (no significant relationship between competitive pressure and perceived usefulness). The structural coefficient of the path between competitive pressure and perceived usefulness (referring to hypothesis 4) is 0.019 ($p < 0.05$). Meanwhile, hypothesis 3 is accepted as the result shows a positive and significant effect between policies on perceived usefulness, with the path coefficient found to be the highest at 0.151 at a significance level of 1% ($p < 0.001$), consistent with the study's expectations.

Moreover, hypothesis 5 and hypothesis 6, perceived ease of use and subjective norms with perceived usefulness, are addressed with the supported data collected. The coefficient of the path between perceived ease of use and perceived usefulness (referring to hypothesis 5) is 0.246 ($p < 0.05$). Hypothesis 6 is accepted as the result shows a positive and significant effect between subjective norms and perceived usefulness; the result shows a positive and significant effect with 0.177 ($p < 0.05$).

Furthermore, hypotheses 7, 8, and 9 have significant and positive effects between perceived usefulness, ease of use, and subjective norms regarding attitude toward use. The coefficient of the path was 0.268, 0.226, and 0.249, respectively ($p < 0.05$), which indicates that hypothesis 7, hypothesis 8, and hypothesis 9 are supported as per the outcomes.

In addition, in hypotheses 10 and 11, the effect of the attitude toward use and behavioural intention to use on actual use is significant and positive. The coefficient of the path between attitude toward use and behavioural intention to use (referring to hypothesis 10) is 0.435 ($p > 0.05$). Further, hypothesis 11 is accepted as a positive and significant relationship between behavioural intention to use and actual use. The result shows a positive and significant relationship of 0.187 ($p < 0.05$).

## 6. Discussion

This study hypothesized relationships based on an integrated model of SCT, TOE, TRA, and TAM to understand the factors affecting the adoption of remote working in Jordanian governmental organizations. Several studies focus on adopting technological systems, such as remote working in governments [118]. Therefore, this study demonstrates a significant relationship between factors and supports the overall hypotheses of the study based on a solid empirical analysis test. Based on the theoretical literature review, this

study presents a set of critical constructs related to the adoption of remote working in Jordanian governmental organizations during the COVID-19 pandemic.

The results confirmed that the organizational factors, namely organizational culture and top management support, significantly and positively affect the perceived usefulness of remote working. H1 and H2 are supported based on this study's empirical results. Organizational culture is the factor that influences other factors in the organizational context, followed by management support as the second most influential factor overall. Results confirmed that accountants realized the positive impact of organizational culture and management support on the perceived usefulness of remote working during the COVID-19 pandemic. Akour, Alshare [53] reported that internal culture and managers' support play a significant role in the employee's perception of the usefulness of technology. These results are also consistent with studies by Davis and Fred [89], Çaldağ, Gökalp [119]. Furthermore, the transition to remote working in governmental organizations during the COVID-19 pandemic helped employees communicate more effectively with management and people [120].

In addition, policies related to remote working positively affect perceived usefulness. At the same time, competitive pressure has an insignificant but positive effect on perceived usefulness. Therefore, H3 is supported while H4 is not supported. According to the path coefficient results, policies positively impact perceived usefulness, followed by competitive pressure as a secondary impact within the environmental context. According to Kim, Jang [121], sound policies released by government or management can increase perceived usefulness for employees, in order for them to understand the system more since they are obligated to use it.

On the other hand, the insignificant result in competitive pressure is consistent with Kuan and Chau [122]. In addition, according to what Alshamaila, Papagiannidis [123] discovered, competitive pressure has an insignificant impact in the cloud computing context. It is assumed that the level of competition between governmental organizations is lower than in private ones [120].

Moreover, study findings show a significant positive impact of perceived ease of use and subjective norms on perceived usefulness. Therefore, H5 and H6 are significantly supported based on the data analysis. In addition, perceived ease of use significantly impacts perceived usefulness, more than subjective norms. In other words, accountants tend to adopt remote working if they better understand the overall benefits and usefulness of working remotely. This outcome is consistent with several previous studies on the behaviour of adopting information systems, such as in Le and Cao [124]. In addition, regarding subjective norms, it was found by Van Acker, Van Buuren [125] that perceived usefulness is affected by the pressures of social influence between workers. Therefore, it is anticipated that perceived ease of use and subjective norms are critical indicators of the perceived usefulness of remote working among governmental accountants. This evidence is supported by studies such as Almaiah, Al-Khasawneh [126].

In addition, study outcomes confirmed a significant positive effect between perceived usefulness, perceived ease of use, and subjective norms on the attitude toward use. Therefore, H7, H8, and H9 are also supported. Accordingly, based on the analyzed data, perceived usefulness has a higher impact on the attitude toward use. In comparison, subjective norms ranked as the second most impacted factor on attitude toward use. In addition, subjective norms impact the perceived ease of use. Several papers are consistent with these outcomes. According to Alshurafat, Al Shbail [26], there is a positive correlation between perceived ease of use and perceived usefulness in adopting online education in the context of Jordanian governmental universities. In addition, the study reported three predictors that indicate the attitude toward use: perceived ease of use, perceived usefulness, and subjective norms, which are consistent with this study.

The outcomes of the study proved that attitude toward use significantly and positively affects the behavioural intention to use. Therefore, H10 is supported, indicating a significant factor directly impacting actual usage. This is similar to the outcomes of Abdekhoda,

Dehnad [92]. Moreover, the outcomes also show a significant positive impact on the behavioural intention to use and the actual use, which indicates that H11 is a supported hypothesis. These outcomes are constant with other findings in the field, such as in Alshurafat, Al Shbail [26].

## 7. Conclusions

This research proposed connections using an integrated model to examine the factors influencing the adoption of remote working in Jordanian governmental organizations. This study shows a strong correlation between factors and supports the overall hypotheses of the study through empirical testing. This study presents a set of key constructs related to the adoption of remote working by Jordanian governmental organizations during the COVID-19 pandemic. The study's implications, limitations, and recommendations are presented in the next subsections.

### 7.1. Implications
#### 7.1.1. Theoretical Implications

This study supports the integration of STC, TOE, TAM, and TRA models as a paradigm to examine factors affecting remote working adoption by Jordanian governmental accountants. All findings and predictions made by this study are empirically tested and justified by previous literature. This study made many theoretical and practical contributions. This contribution begins by providing new motivation for existing accounting research to broaden its theoretical lens to include comprehensive remote working adoption factors. The results indicate that the factors adopted in this study influence remote working during COVID-19. Therefore, the results expand our awareness of the adoption process.

#### 7.1.2. Contextual Implications

The findings of this study have a wide range of implications, many of which are especially appropriate to Jordan. The implications of the study's findings can likewise be applied to other countries. The study's findings may interest various stakeholders, including policymakers, investors, managers, and academics. Moreover, the study's findings are essential, even in other developing contexts, for developing a consistent set of requirements to improve remote working. This study offers important implications for Jordanian governmental organizations. When determining factors influencing working remotely, organizations can decide whether to adopt this work [26].

#### 7.1.3. Managerial Implications

Jordanian governmental organizations might improve the quality of remote accounting work. The quality enhancement will improve workers' control and risk management, leading to a new perception of the benefits and ease of remote work. In addition, managers are encouraged to identify the benefits of transitioning to remote working for accountants. Employees ought to understand more about the system of remote working by educating themselves about how to use it to facilitate their work [127].

In addition, the Jordanian government and policymakers can establish laws and policies to regulate remote working with regulations that ease its use and adoption within governmental organizations. Governmental laws and financial support can help to implement training and courses to develop workers' information technology skills [128].

### 7.2. Limitations and Future Research Recommendations

This study was limited to the Jordanian context based on the perceptions of accountants who worked in governmental organizations during the COVID-19 pandemic. In addition, the study's results and outcomes target accountants, making the outcomes ungeneralizable to other jobs within other organizations. Lastly, the research model encompasses limited factors from other research and from previous literature.

This study has focused on many factors adopted from different models and theories. However, this study has excluded the technological context from the TOE model, where it is recommended that future research expand the model by adopting the technological context. In addition, further research can implement the study over new geographical contexts in developed and developing countries to compare outcomes. Future researchers could conduct studies in new contexts for specific jobs such as managerial accountants, tax accountants, or roll payment accountants, which will widen their insights into the objectives of this study. This study is from both individual and organizational level perspectives. Further research should focus on one level to gain more details based on a single perspective.

**Author Contributions:** Conceptualization, Q.A.O., M.O.A.S. and H.A.A.; Methodology, H.A. (Husam Ananzeh); Validation, H.A. (Hashem Alshurafat); Formal analysis, H.A. (Husam Ananzeh); Investigation, M.O.A.S. and H.A. (Husam Ananzeh); Resources, M.O.A.S.; Data curation, H.A. (Hashem Alshurafat); Writing—original draft, Q.A.O.; Writing—review & editing, H.A. (Hashem Alshurafat), M.O.A.S., H.A. (Husam Ananzeh) and H.A.A.; Visualization, H.A.A.; Supervision, Q.A.O. and H.A.A.; Project administration, Q.A.O. and H.A. (Hashem Alshurafat); Funding acquisition, H.A.A. All authors have read and agreed to the published version of the manuscript.

**Funding:** Open access funding provided by Qatar National Library.

**Institutional Review Board Statement:** Not applicable.

**Informed Consent Statement:** Not applicable.

**Data Availability Statement:** Not applicable.

**Conflicts of Interest:** On behalf of all authors, the corresponding author states that there are no conflict of interest.

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
