# Peer review of "Factors Affecting Accountants’ Adoption of Remote Working: Evidence from Jordanian Governmental Organizations"

_sustainability, doi:10.3390/su151713224_

Round 1
Reviewer 1 Report
Dear authors,
Thank you for your manuscript.
My comments are:
The literature review is rather superficial. I suggest that the authors also review the theories utilised to investigate this acceptance.
I urge the author integrate the present research gaps in TAM within the research model. This is necessary to confirm that the writers addressed a large knowledge gap. Please evaluate the following systematic review studies in order to find the appropriate gap:
Marangunić, N., Granić, A. Technology acceptance model: a literature review from 1986 to 2013. Univ Access Inf Soc 14, 81–95 (2015)
Granić, A., Marangunić, N. Technology acceptance model in educational context: A systematic literature review. British Journal of Educational Technology, 50(5), pp. 2572-2593. https://doi.org/10.1111/bjet.12864
Rosli, M.S.; Saleh, N.S.; Md. Ali, A.; Abu Bakar, S.; Mohd Tahir, L. A Systematic Review of the Technology Acceptance Model for the Sustainability of Higher Education during the COVID-19 Pandemic and Identified Research Gaps. Sustainability 2022, 14, 11389. https://doi.org/10.3390/su141811389
3.6 Subjective Norms - Reiterate the TRA-derived definition that serves as the core concept of SN in TAM.
Numerous TAM studies disregarded the role of attitude as a mediator of BI. The writers must expound on this to ensure that their study is comparable to and in line with that of other scholars. You might refer to the following publication regarding the necessity to include or exclude attitude in TAM:
Fazil Abdullah, Rupert Ward, Ejaz Ahmed, Investigating the influence of the most commonly used external variables of TAM on students’ Perceived Ease of Use (PEOU) and Perceived Usefulness (PU) of e-portfolios, Computers in Human Behavior, Volume 63, 2016, Pages 75-90, https://doi.org/10.1016/j.chb.2016.05.014.
What is the recommended minimum sample size? How do the authors determine the sample size minimum? Does this study meet the minimum needed sample size? Please provide good justification and evidence from the literature for this claim.
The research instrument requires additional clarification. Please specify the origin of the items and the quantity involved. How many were altered or adopted from another scholar? Cite the original author.
Why was CB-SEM ruled out and not utilised in this research? As CB-SEM is more appropriate for TAM since it examines current theory, but PLS-SEM is more appropriate for developing new theory.
6. Conclusion, implications, recommendation - This should be the subtopic of the discussion. The discussion is excessively brief and devoid of analysis.
The conclusion should appear at the end of the manuscript.
Author Response
|
Comments |
Responses |
|
Reviewer 1 |
|
|
Dear authors,
Thank you for your manuscript.
My comments are: |
Thank you for your comments. We have addressed all the comments. |
|
The literature review is rather superficial. I suggest that the authors also review the theories utilised to investigate this acceptance. |
This comment has been met by reviewing more relevant studies. |
|
I urge the author integrate the present research gaps in TAM within the research model. This is necessary to confirm that the writers addressed a large knowledge gap. Please evaluate the following systematic review studies in order to find the appropriate gap:
Marangunić, N., Granić, A. Technology acceptance model: a literature review from 1986 to 2013. Univ Access Inf Soc 14, 81–95 (2015)
Granić, A., Marangunić, N. Technology acceptance model in educational context: A systematic literature review. British Journal of Educational Technology, 50(5), pp. 2572-2593. https://doi.org/10.1111/bjet.12864
Rosli, M.S.; Saleh, N.S.; Md. Ali, A.; Abu Bakar, S.; Mohd Tahir, L. A Systematic Review of the Technology Acceptance Model for the Sustainability of Higher Education during the COVID-19 Pandemic and Identified Research Gaps. Sustainability 2022, 14, 11389. https://doi.org/10.3390/su141811389 |
This comment has been met by reviewing more relevant studies. All the suggested papers have been reviewed and added to the literature section. |
|
3.6 Subjective Norms - Reiterate the TRA-derived definition that serves as the core concept of SN in TAM. |
The required reiteration has been provided. |
|
Numerous TAM studies disregarded the role of attitude as a mediator of BI. The writers must expound on this to ensure that their study is comparable to and in line with that of other scholars. You might refer to the following publication regarding the necessity to include or exclude attitude in TAM: Fazil Abdullah, Rupert Ward, Ejaz Ahmed, Investigating the influence of the most commonly used external variables of TAM on students’ Perceived Ease of Use (PEOU) and Perceived Usefulness (PU) of e-portfolios, Computers in Human Behavior, Volume 63, 2016, Pages 75-90, https://doi.org/10.1016/j.chb.2016.05.014. |
The required justification has been provided. |
|
What is the recommended minimum sample size? How do the authors determine the sample size minimum? Does this study meet the minimum needed sample size? Please provide good justification and evidence from the literature for this claim. |
The required justification has been provided. |
|
The research instrument requires additional clarification. Please specify the origin of the items and the quantity involved. How many were altered or adopted from another scholar? Cite the original author. |
The required clarification has been provided. |
|
Why was CB-SEM ruled out and not utilised in this research? As CB-SEM is more appropriate for TAM since it examines current theory, but PLS-SEM is more appropriate for developing new theory. |
The required clarification has been provided. |
|
6. Conclusion, implications, recommendation - This should be the subtopic of the discussion. The discussion is excessively brief and devoid of analysis. |
The paper has been restructured to meet this comment. Thank you. |
|
The conclusion should appear at the end of the manuscript. |
The paper has been restructured to meet this comment. Thank you. |
Reviewer 2 Report
Very interesting article. The method of analysis is presented in great detail.
Authors should complete and correct the article as follows:
1. The purpose of the study should be clearly stated in the abstract and the introduction. The authors refer to the objective in lines 282-284, 589, and the reader does not find the study goals specified in the article.
2. The same applies to research questions. The reference to them is in lines 282-284, and from the article's content, it can be concluded that these are questions of the questionnaire. I point out to the authors that it is not the same. Line 63 mentions one question (“Therefore, this study answers the following question”) and lines 282-284 talk about “research questions”
3. For the statement “Several studies have dealt with the technology acceptance…” on line 49, it is necessary to provide references from the literature so that the reader can find these studies.
4. In line 188 is "Competitive pressure and the perceived usefulness". Is this a subsection title or a subsection title?
5. In lines 468, 470, 473, 477, and 480 for Hypotheses 5-11, it should be p<0.05 and not as written by the authors p>0.05
Author Response
|
Reviewer 2 |
|
|
Very interesting article. The method of analysis is presented in great detail.
Authors should complete and correct the article as follows: |
Thank you for your comments. We have addressed all the comments. |
|
1. The purpose of the study should be clearly stated in the abstract and the introduction. The authors refer to the objective in lines 282-284, 589, and the reader does not find the study goals specified in the article. |
The purpose of this paper has been clarified in the abstract and the introduction. |
|
2. The same applies to research questions. The reference to them is in lines 282-284, and from the article's content, it can be concluded that these are questions of the questionnaire. I point out to the authors that it is not the same. Line 63 mentions one question (“Therefore, this study answers the following question”) and lines 282-284 talk about “research questions” |
Claims at lines 282-284 were very distributive, we have abounded some of these claims to make the paragraph understandable. Thank you for your comment. |
|
3. For the statement “Several studies have dealt with the technology acceptance…” on line 49, it is necessary to provide references from the literature so that the reader can find these studies. |
Citations have been provided.
|
|
4. In line 188 is "Competitive pressure and the perceived usefulness". Is this a subsection title or a subsection title? |
The paper has been restructured to meet this comment. Thank you. |
|
5. In lines 468, 470, 473, 477, and 480 for Hypotheses 5-11, it should be p<0.05 and not as written by the authors p>0.05 |
This comment has been met, thank you. |

Reviewer 3 Report
Thank you for the opportunity to review this paper. I think the authors have conducted an interesting piece of research on an important topic. I have some comments that I hope they find useful:
In the introduction, I would suggest you briefly include a paragraph indicating the findings and contributions.
Section 2 seems short. I would recommend adding more content, and if this is not possible, to merge sections 2 and 3.
Did you control for common method bias?
And for early vs late respondent bias, and non-response bias?
Were all the respondents native in the language of the survey?
I would recommend merging sections 6.2 and 6.3
Good luck!
Author Response
|
Reviewer 3 |
|
|
Thank you for the opportunity to review this paper. I think the authors have conducted an interesting piece of research on an important topic. I have some comments that I hope they find useful: |
Thank you for your comments. We have addressed all the comments. |
|
In the introduction, I would suggest you briefly include a paragraph indicating the findings and contributions. |
A new paragraph has been added |
|
Section 2 seems short. I would recommend adding more content, and if this is not possible, to merge sections 2 and 3. |
This comment has been met by reviewing more relevant studies. |
|
Did you control for common method bias? And for early vs late respondent bias, and non-response bias? |
A common method bias test was added. |
|
Were all the respondents native in the language of the survey? |
Yes, and this has been added to the paper |
|
I would recommend merging sections 6.2 and 6.3 |
The paper has been restructured to meet this comment. Thank you. |
|
Good luck! |
Thank you so much. |

Reviewer 4 Report
Please, better explain the sentence at p. 2, line 47-48 : "Still, it was a temporary policy to work that surprised account- 47 ants remotely without any plans or preparations for such circumstances."
p.2, line 58: "This study examined the accountants' perceptions". Perceptions of what? please explain.
The literature review on remore work is poorly developed; there are also literature review that might be useful in this respect. Besides, authors should be consider the difference between the pros and cons of remote work during the pandemic and the remote work before the pandemic. This should be considered when discussing the findings, but also in the presentation of the theoretical basis as well as the hypotheses, since the shift from work in the office and remore work was actually pushed suddenly by the pandemic, and was not an organizational choice.
Authors should explain why their study is needed, not only what it adds to the previous literature.
How their findings might be useful for other countries or workers?
"H3: Policies positively impact the perceived usefulness of using remote working for 186 accountants within the Jordanian governmental organization." Which specific policies are the authors referring to?
Authors must deliver all the relevant information about the instruments used; no real information is offered on the actual scales used to collect the data. Which were? were they validated?
In the light of this, it is impossible to assess the quality of the data collected and the findings
English needs to be brushed up
Author Response
|
Reviewer 4 |
|
|
Please, better explain the sentence at p. 2, line 47-48 : "Still, it was a temporary policy to work that surprised account- 47 ants remotely without any plans or preparations for such circumstances." |
We have abounded this claim as it confuses readers. |
|
p.2, line 58: "This study examined the accountants' perceptions". Perceptions of what? please explain. |
More elaboration has been provided. “This study examined the accountants' perceptions about the factors that impact their adoption of remote working within the Jordanian governmental organizations.” |
|
The literature review on remore work is poorly developed; there are also literature review that might be useful in this respect. Besides, authors should be consider the difference between the pros and cons of remote work during the pandemic and the remote work before the pandemic. This should be considered when discussing the findings, but also in the presentation of the theoretical basis as well as the hypotheses, since the shift from work in the office and remore work was actually pushed suddenly by the pandemic, and was not an organizational choice. |
This comment has been met by reviewing more relevant studies. |
|
Authors should explain why their study is needed, not only what it adds to the previous literature. How their findings might be useful for other countries or workers? |
The required paragraph has been added please refer to the penultimate paragraph at the introduction section. |
|
"H3: Policies positively impact the perceived usefulness of using remote working for 186 accountants within the Jordanian governmental organization." Which specific policies are the authors referring to? |
The required clarification has been provided. |
|
Authors must deliver all the relevant information about the instruments used; no real information is offered on the actual scales used to collect the data. Which were? were they validated? In the light of this, it is impossible to assess the quality of the data collected and the findings |
The required clarification has been provided. |
|
English needs to be brushed up |
Done. |

Reviewer 5 Report
The presented work is a kind request to fill out the questionnaire but not the research article. It can not be evaluated or reviewed as such a paper.
Author Response
|
Reviewer 5 |
|
|
The presented work is a kind request to fill out the questionnaire but not the research article. It can not be evaluated or reviewed as such a paper. |
Thank you. |
|
Academic Editor Notes |
|
|
Dear authors, I think your paper needs to be improved considerably. Particularly: |
Thank you for your comments. We have addressed all the comments. |
|
- It is not possible to clearly understand what is the objective of the research or what the authors want to add to the existing literature. |
The required clarification has been provided. |
|
- The literature section is very weak, and needs important additions, including, in addition to those reported by the reviewers: - Nagy, J.T. (2018). Evaluation of online video usage and learning satisfaction: An extension of the technology acceptance model. International Review of Research in Open & Distance Learning, 19(1), 160–184. - Persico, D., Manca, S., & Pozzi, F. (2014). Adapting the technology acceptance model to evaluate the innovative potential of e-learning systems. Computers in Human Behavior, 30, 614–622. - Murillo, G. G., Novoa-Hernández, P., & Rodriguez, R. S. (2021). Technology Acceptance Model and Moodle: A systematic mapping study. Information Development, 37(4), 617-632. |
This comment has been met by reviewing more relevant studies. All the suggested papers have been reviewed and added to the literature section. |
|
- As correctly highlighted by a reviewer, authors should be considering that remote working was not an organizational choice, but an obligatory solution following the pandemic. This should be considered when discussing research findings. |
This comment has been met. |
|
- The discussion of the results is very poor, it doesn't seem to explain well what has been done in the course of the work. |
The paper has been restructured to meet this comment. Thank you. |
|
- The paper necessarily needs proofreading. |
Done. |

Round 2
Reviewer 3 Report
I have no further comments
Author Response
Thank you so much

Reviewer 4 Report
In my previous review I asked for deliveingr all the relevant information about the instruments used; no real information is offered on the actual scales used to collect the data. Which were? were they validated?
Authors did not address this point. This means that it is impossible to assess the quality of the data collected and thereffre of theor findings
Author Response
|
The required explanations on the validity of the research instrument were added in the second paragraph of section 4.2. |

Reviewer 5 Report
The authors analyzed a popular topic about remote work. Their research's specificity is the country and selected respondents (accountants). After careful reading here are some suggestions for improvement:
a) Description of hypothesis could reveal why this dimension is chosen for analyses. The authors describe every dimension as already analyzed in science but do not reveal why they as researchers choose once again to analyse those dimensions.
b) Deeper explanation of why 155 respondents as a sample size are enough for valid research results should be provided.
c) Paragraphs from 361-372 lines would be recommended to move to the 4.2 subchapter.
d) 407-416 lines as well as 433-434 lines should be placed on the reference list but not in the text.
e) Please check the direction of the arrows in figure 1 compared with figure 2. It seems that direction of the arrows is different.
I would suggest to the authors clearly define the aim of the research and write the article keeping the direction of solving this defined aim. The authors have analyzed a lot and presented data, but the article seems not united among some subchapters and paragraphs.
Author Response
|
The authors analyzed a popular topic about remote work. Their research's specificity is the country and selected respondents (accountants). After careful reading here are some suggestions for improvement: |
Thank you so much! |
|
a) Description of hypothesis could reveal why this dimension is chosen for analyses. The authors describe every dimension as already analyzed in science but do not reveal why they as researchers choose once again to analyse those dimensions. |
The required explanations are provided in the second and third paragraphs of section 3.1. |
|
b) Deeper explanation of why 155 respondents as a sample size are enough for valid research results should be provided. |
The required explanations are provided in the third, fourth and fifth paragraphs of section 3.1. |
|
c) Paragraphs from 361-372 lines would be recommended to move to the 4.2 subchapter. |
Done. |
|
d) 407-416 lines as well as 433-434 lines should be placed on the reference list but not in the text. |
Done. |
|
e) Please check the direction of the arrows in figure 1 compared with figure 2. It seems that direction of the arrows is different. |
The arrows are correct. |
|
I would suggest to the authors clearly define the aim of the research and write the article keeping the direction of solving this defined aim. The authors have analyzed a lot and presented data, but the article seems not united among some subchapters and paragraphs. |
The aim if this study is well-defined at the end of the second paragraph of this introduction section. The authors make sure that this aim is consistent among the paper sections. |
